# Topologically ordered time crystals

Thorsten B. Wahl [1], Bo Han [2,3] & Benjamin Béri [1,3] ✉

Time crystals are a dynamical phase of periodically driven quantum many-body systems where discrete time-translation symmetry is broken spontaneously. Time-crystallinity however subtly requires also spatial order, ordinarily related to further symmetries, such as spin-flip symmetry when the spatial order is ferromagnetic. Here we define topologically ordered time crystals, a time-crystalline phase borne out of intrinsic topological order—a particularly robust form of spatial order that requires no symmetry. We show that many-body localization can stabilize this phase against generic perturbations and establish some of its key features and signatures, including a dynamical, time-crystal form of the perimeter law for topological order. We link topologically ordered and ordinary time crystals through three complementary perspectives: higher-form symmetries, quantum error-correcting codes, and a holographic correspondence. Topologically ordered time crystals may be realized in programmable quantum devices, as we illustrate for the Google Sycamore processor.

Time Crystals (TCs)[1–3] are periodically driven (i.e., Floquet) systems that spontaneously break discrete time-translation symmetry and a global internal symmetry $G$ (such as the spin-flip symmetry of an Ising ferromagnet)[4–10]. To form a phase of matter, TCs must be protected from the drive-induced heating. A route is via many-body localization[11–13] (MBL): with strong disorder, local degrees of freedom emerge that remain inert if their energies are much below the driving frequency $2\pi/T$[4–10]. TCs display long-range spatiotemporal order[7,8]. That is, local order parameters $O_j$ (i.e., operators at position $j$ that transform nontrivially under $G$) exist such that, for large $|j − k|$ and for any eigenstate $|\psi\rangle$ of the time-evolution operator, the time-dependent expectation $\langle\psi|O_j(mT)O_k(0)|\psi\rangle$ has period greater than one as the function of the integer $m$.

Many-body systems can however display orders other than symmetry breaking. A key example is topological order (TO), with anyonic quasiparticles or ground-state degeneracy depending only on the topology of the configuration space[14–16]. These striking phenomena motivated numerous advances (e.g., symmetry-protected or -enriched topological phases[17–19]) and applications [e.g., quantum error correcting (QEC) codes[14,20,21]]. Studies of topological matter (although mainly with topology distinct from TO) also include dynamical quenches[22,23] or driven systems [24–28].

In this work, we ask: how may TCs emerge from TO as the underlying spatial quantum fabric? TO has no global symmetry $G$ in the usual sense, has no local order parameters, and its nonlocal features may seem in tension with MBL. Hence the resulting topologically ordered TCs (TTCs) would be fundamentally distinct from TCs (or Floquet symmetry-protected topological phases[25–28]), but even their definition seems challenging.

A fruitful analogy to TCs does however arise[7,8] if we generalize $G$ to higher-form symmetries[29–31]: TO appears as the spontaneous breaking of these, and this does have—albeit nonlocal—order parameters. Here we define TTCs based on these ingredients.

We also show that TTCs form a phase of matter: they do not need fine-tuning and have observable signatures. We establish this by invoking MBL. As TO requires two dimensions (2D) or higher, TTCs require MBL in 2D or above, and a form of TO compatible with MBL: so-called nonchiral Abelian TO[14–16,32–35]. In 2D, MBL may arise as a long-lived pre-thermal phase persisting beyond current experimental time-scales[36–39]. Focusing on this pre-thermal regime, we show that TTCs are robust against perturbations, and we establish some key TTC features. These include a dynamical, time-crystal form of the perimeter law, a TO signature—via nonlocal order parameters—of the same status as long-range correlations for spontaneous symmetry breaking[32,33,40,41]. A recent experiment realizing our proposed TTC on an intermediate scale quantum processor supports the predictions accessible at that scale[42].

To establish our results, we formulate TTCs as Floquet-MBL versions of TO systems related to QEC codes. On the technical level, this

[1]DAMTP, University of Cambridge, Cambridge, UK. [2]Department of Condensed Matter Physics, Weizmann Institute of Science, Rehovot, Israel. [3]T.C.M. Group, Cavendish Laboratory, University of Cambridge, Cambridge, UK. ✉e-mail: bfb26@cam.ac.uk

allows us to use a topological variant[35] of local integrals of motion (LIOMs)[43–47], an MBL framework that has been key for ordinary TCs[5,7,8]. Such topological LIOMs (tLIOMs) will give a similarly useful framework for TTCs. Conceptually, our approach highlights QEC codes as another unifying perspective bridging TCs and TTCs. We illustrate our findings, including both the QEC and the higher-form symmetry perspectives, on the surface code[14,20,21] (the simplest TO); using these systems we also highlight yet another unifying view: a holographic TTC–to–TC correspondence. We also explain how these surface-code-based (prethermal) TTCs can be created in the Google Sycamore processor[48,49] from available ingredients.

## Results

### TTCs via QEC codes and generalized symmetries

We seek TTCs in Floquet systems. Over a driving period $T$, the time evolution is generated by the Floquet unitary $U_F \equiv \mathcal{T} \exp[-i \int_0^T H(t)dt]$, where $\mathcal{T}$ is time ordering and $H(t)$ is a local Hamiltonian (i.e., a sum of finite-range bounded-norm terms[50]) at time $t$. In particular, $O(mT) = U_F^{m\dagger} O(0) U_F^m$ for any operator $O = O(0)$. We mostly focus on two-step drives, i.e., $H(t) = H_0/t_0$ for $0 \leqslant t < t_0$ and $H(t) = H_1/t_1$ for $t_0 \leqslant t < t_0 + t_1 \equiv T$. This gives $U_F = \exp(-iH_1) \exp(-iH_0)$.

For simplicity, we consider lattice systems of $N$ qubits, but the ideas are more general, as we shall discuss. To explain how TTCs arise, we first describe $U_F$ for ordinary TCs, focusing on 1D systems with $G = \mathbb{Z}_2$ (i.e., spin-flip) symmetry[4,5,7,8,25]. The symmetry operator is $P = \prod_j X_j$ and, by $P^\dagger Z_j P = -Z_j$, we can use $Z_j$ as a local order parameter ($X_j$ and $Z_j$ are Pauli $X$ and $Z$ operators at site $j$, respectively). The key TC features originate from the unperturbed limit where $H_0 = \sum_j J_j Z_j Z_{j+1}$, with $J_j$ real, and $H_1 = \frac{\pi}{2} \sum_j X_j$. In this limit, the Floquet unitary is, up to a phase,

$$U_{F0}^{(TC)} = P \exp(-iH_0). \tag{1}$$

The operators $S_j \equiv Z_j Z_{j+1}$ ($j = 1, \ldots N-1$) and $P$ provide a complete set of integrals of motion; their eigenvalues $s_j, p \in \{-1, 1\}$ fully specify the eigenbasis $|\mathbf{s}, p\rangle$ of $U_{F0}^{(TC)}$, i.e., the Floquet eigenstates. These are spin-glass eigenstates ($s_j = -1$ marks a domain wall between sites $j$ and $j+1$); since $\langle \mathbf{s}, p | Z_j Z_k | \mathbf{s}, p \rangle$ is long-ranged, $\mathbb{Z}_2$ symmetry is spontaneously broken. The role of $H_0$ in Eq. (1) is to imprint this symmetry breaking. The prefactor $P$, in turn, introduces period-doubling: by $P^\dagger Z_j P = -Z_j$ and $[Z_j, H_0] = 0$, we have $Z_j(mT) = (-1)^m Z_j(0)$ and thus the system has long-range spatiotemporal order.

Strong disorder in $J_j$ makes $U_{F0}^{(TC)}$ MBL. This allows one to argue that the TC is robust against perturbing $U_{F0}^{(TC)}$ by local terms in $H_{0,1}$[7,8].

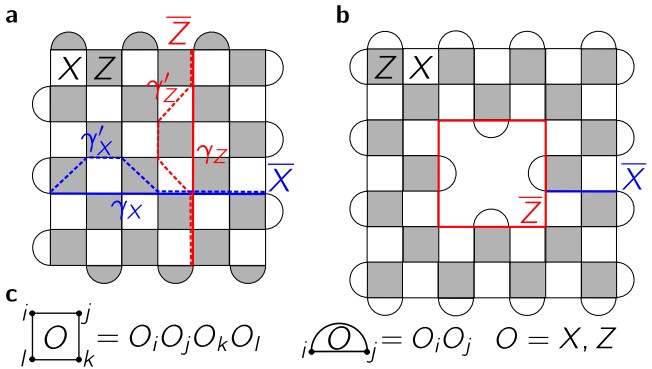

**Fig. 1 | Surface codes furnishing a single logical qubit.** In both **a** and **b**, qubits are at the vertices and the check operators are $S_P = \prod_{i \in P} O_i$, with $O = X, Z$, depending on the plaquette $P$, cf. **c** The logical operators $\overline{O} = \mathcal{O}_{\gamma_O} = \prod_{i \in \gamma_O} O_i$ (with $O = X, Z$) are Pauli strings along noncontractible paths $\gamma_O$, connecting boundaries with $O$-type $S_P$ (**a**) and/or encircling one with opposite type $S_P$ (**b**). By $O^2 = \mathbb{1}$, $\mathcal{O}_{\gamma_O}$ can be deformed into $\mathcal{O}_{\gamma_O'}$ via $O$-type checks (cf. **a**, dashed).

Remarkably, this holds even for $\mathbb{Z}_2$ symmetry breaking terms, a feature dubbed absolute stability[7]. TTCs will enjoy a similarly enhanced robustness, even compared to TO.

To prepare for constructing TTCs, we re-interpret $U_{F0}^{(TC)}$ via QEC[51,52] (see also Ref. 53). In this language, $S_j$ are the check operators for the $N$-qubit repetition code: measuring them allows detecting a change in $\mathbf{s}$ to detect if bit-flips (effected by $X_i$) have occurred. $P$ and $Z_j$, in turn, are conjugate logical Pauli operators: each preserves $\mathbf{s}$, but they anticommute and enact nontrivial operations within the doublet $|\mathbf{s}, p\rangle$, i.e., the logical qubit. (Any of the $Z_j$ is an equally valid logical operator choice since $Z_i \propto Z_j \prod_{k \in V} S_k$ for some set $V$.)

We now use this QEC perspective to construct TTCs. The key observation is that the unperturbed limit of a broad family of TO states —namely, nonchiral, Abelian TOs—also arise as common eigenspaces of suitable mutually commuting and local check operators $S_P$[14–16]. For simplicity, we will mostly assume that, similarly to the TC example, the $S_P$ and the logical operators are Pauli strings (tensor products of Pauli operators on qubits); while this captures only a subset of TOs, it already conveys the essential ideas.

The simplest example is the surface code[14,20,21], cf. Fig. 1. Here $S_P$ are Pauli strings around plaquettes $P$. For simplicity, we focus on systems furnishing a single logical qubit (i.e., two-fold spectral degeneracies). We choose $\overline{X} = \mathcal{X}_{\gamma_X}$ for the logical Pauli $X$ with $\mathcal{X}_{\gamma_X} = \prod_{j \in \gamma_X} X_j$ along path $\gamma_X$. By multiplying $\mathcal{X}_{\gamma_X}$ with adjacent $S_P = \prod_{j \in P} X_j$, one may deform $\mathcal{X}_{\gamma_X}$ into an equally valid $\overline{X}$ choice $\mathcal{X}_{\gamma_X'}$ along path $\gamma_X'$; as deformations cannot detach these paths from their respective surface-code boundaries, the paths are noncontractible. Similar observations hold for $\overline{Z} = \mathcal{Z}_{\gamma_Z} = \prod_{j \in \gamma_Z} Z_j$.

The above illustrate key features that we shall use. They hold for any nonchiral Abelian TO[14–16] where again logical operators commute with all $S_P$, run along noncontractible paths that are deformable by check-operator products, and paths for conjugate pairs (such as $\overline{X}$ and $\overline{Z}$) can be chosen to intersect only once. These all have their anyon interpretation[14–16]: the $S_P$ eigenvalues label anyon configurations, a logical operator $\mathcal{W}_\gamma$ drags a corresponding anyon along its path $\gamma$, and its algebra $\overline{O}^\dagger \mathcal{W}_\gamma \overline{O} = e^{2i\theta_{OW}} \mathcal{W}_\gamma$ with its conjugate $\overline{O}$ encodes mutual statistics through the angle $\theta_{OW}$.

The surface code drive

$$U_{F0} = \overline{O} \exp(-iH_0), \quad H_0 = \sum_P J_P S_P, \tag{2}$$

with $J_P$ real, has structure identical to Eq. (1), suggesting that it exemplifies TTCs in their unperturbed limit. (See also Refs. 7,8,54 for similar constructions.) The Floquet eigenstates $|\mathbf{s}, o\rangle$ now form doublets labeled by the $\overline{O}$ eigenvalue $o$. By its role being akin to $P$ in TCs, we view $\overline{O}$ as a generalized symmetry[29–31]. Taking $\overline{O} = \mathcal{X}_{\gamma_X}$ for concreteness, by $\overline{O}^\dagger \mathcal{Z}_{\gamma_Z} \overline{O} = -\mathcal{Z}_{\gamma_Z}$ we can then view the conjugate logical $\mathcal{Z}_{\gamma_Z}$ as order parameter. Unlike $Z_j$ for the TC, $\mathcal{Z}_{\gamma_Z}$ is not local, but has 1D support. (Hence $\overline{O}$, acting on 1D charged objects, is a 1-form symmetry[30,31].) $\mathcal{Z}_{\gamma_Z}$ is however local transversally to $\gamma_Z$. It is thus meaningful to consider $\langle \mathbf{s}, o | \mathcal{Z}_{\gamma_Z} \mathcal{Z}_{\gamma_Z'} | \mathbf{s}, o \rangle$ at large separation $d(\gamma_Z, \gamma_Z') \gg 1$. In this sense, the TO states $|\mathbf{s}, o\rangle$ are long-range ordered. By $\overline{O}^\dagger \mathcal{Z}_{\gamma_Z} \overline{O} = -\mathcal{Z}_{\gamma_Z}$ and $[\mathcal{Z}_{\gamma_Z}, H_0] = 0$, we have

$$\mathcal{Z}_{\gamma_Z}(mT) = (-1)^m \mathcal{Z}_{\gamma_Z}(0). \tag{3}$$

The drive $U_{F0}$ thus has spatiotemporal features similar to a TC, but generalized to TO via 1-form symmetries. [Drives with symmetry $\overline{O} = \mathcal{Z}_{\gamma_Z}$ or $\overline{O} = i\mathcal{Z}_{\gamma_Z} \mathcal{X}_{\gamma_X}$, with order parameter $\mathcal{X}_{\gamma_X}$, would be equally valid.] Arising from $\overline{O}^\dagger \mathcal{Z}_{\gamma_Z} \overline{O} = e^{2i\theta_{OZ}} \mathcal{Z}_{\gamma_Z} = -\mathcal{Z}_{\gamma_Z}$, Eq. (3) originates in surface-code anyons' mutual semion statistics.

The path choice in $\overline{O} = \mathcal{X}_{\gamma_X}$, or $H_1 = \frac{\pi}{2} \sum_{j \in \gamma_X} X_j$, is arbitrary: any of $\gamma_X$'s deformations gives a valid $U_{F0}$. One may also use a product over

paths, $\overline{O} = \prod_{\gamma_x \in \Gamma} \mathcal{X}_{\gamma_x}$ with any $\Gamma$ of odd cardinality; then $\overline{O}$ anti-commutes with $\overline{Z}$ (and commutes with each $S_P$) hence is a valid $\overline{X}$. If, as in Fig. 1a, (i) the $S_P$ are purely $X$- or $Z$-strings (they form a Calderbank-Shor-Steane code[52]), (ii) each $Z$-check features an even number of $Z_j$, and (iii) the code has $\overline{Z} = \mathcal{Z}_{\gamma_Z}$ with odd-length $\gamma_Z$, then the uniform, and thus path-choice-free, $H_1 = \frac{\pi}{2}\sum_j X_j$ translates to such $\overline{O}$ because $\exp(-iH_1) \propto \prod_j X_j$ commutes with each $S_P$ but anticommutes with $\overline{Z}$.

The features of $U_{FO}$ motivate the following:

**Definition 1.** $U_F = \exp(-iH_1)\exp(-iH_0)$ is a TTC if (1) it has TO in all its eigenstates $|\psi\rangle$; (2) a (smeared) logical operator $\widetilde{\mathcal{W}}_\gamma$, along path $\gamma$, exists such that, for any $|\psi\rangle$ and up to corrections exponentially small in the linear system size, $\langle\psi|\widetilde{\mathcal{W}}_\gamma(mT)\widetilde{\mathcal{W}}_\gamma(0)|\psi\rangle$ has finite period greater than one as the function of the integer $m$; (3) these features are robust against local perturbations in $H_{0,1}$.

As we shall explain, perturbations smear logical operators (and the $S_P$) of TO MBL systems over a lengthscale set by the localization length $\xi$[35]. The anyon interpretation, above Eq. (2), now holds for these smeared operators, and it is this smearing that we allow (and indicate by tilde) in the definition. (We have $\xi = 0$ for $U_{FO}$.) As TO eigenstates imply that $\widetilde{\mathcal{W}}_\gamma$ is deformable to $\widetilde{\mathcal{W}}_{\gamma'}$, the definition replaces (generalized) long-range order by single-path expectations and the separate requirement of eigenstate TO. By the TO in $|\mathbf{s}, o\rangle$ and the period doubling in Eq. (3), properties (1) and (2) are satisfied by $U_{FO}$ in Eq. (2) for $\widetilde{\mathcal{W}}_\gamma = \mathcal{Z}_{\gamma_Z}$ and $|\psi\rangle = |\mathbf{s}, o\rangle$. We require (3) to capture a phase of matter.

## Robustness of TTCs via MBL

We next show that the robustness of TO MBL implies that $U_{FO}$, and its generalizations to other nonchiral Abelian TO (upon suitably replacing $S_P$ and $\overline{O}$), satisfy (3) in Definition 1. We use the QEC language to treat conventional and TO MBL on the same footing[35]. A key concept is that of a local unitary $\widetilde{U}$, i.e., a finite-time evolution with a local Hamiltonian[17,50]. If $O_i$ is local to site $i$, then $\widetilde{O}_i = \widetilde{U}O_i\widetilde{U}^\dagger$ is quasilocal: in its Pauli-string expansion, operators supported at distance $\ell$ from $i$ have coefficients decaying exponentially with $\ell$.

A system is MBL if the following holds robustly against local perturbations[35,43–47]: the eigenbasis $|\psi_{\mathbf{s},\mathbf{o}}\rangle$ of $U_F$ satisfies $|\psi_{\mathbf{s},\mathbf{o}}\rangle = \widetilde{U}|\mathbf{s}, \mathbf{o}\rangle$ where $|\mathbf{s}, \mathbf{o}\rangle$ are eigenstates of $N - k$ local operators $S_P$ (with eigenvalues in vector $\mathbf{s}$) and $k$ logical operators $\overline{O}_j(\mathbf{s})$ [with eigenvalues in $\mathbf{o}$, $j \in \{1, ..., k\}$, and $k$ being $N$-independent] and $\widetilde{U}$ is a local unitary (set by the details of $U_F$). The $\mathbf{s}$ dependence in $\overline{O}_j(\mathbf{s})$ arises when it is perturbations that split the logical-subspace degeneracies. As in our examples above, we shall mostly focus on $k = 1$.

For $U_F = \exp(-iH)$ with static 2D Hamiltonian $H = \sum_P J_P S_P + \delta H$, with $S_P$ for a TO and $J_P$ disordered with distribution of width $\delta J$, (pre-thermal) TO MBL is expected for any $\delta H$ with local terms with couplings of order $g \ll \delta J$[32–35]. (See also Ref. 55 for numerics.) TO MBL is robust in this sense. It is pre-thermal because perturbative arguments[39], adapted to TO, suggest that its MBL has thermalization timescale $t_{th}$ longer than exponential in $\delta J/g$. Beyond $t_{th}$, MBL breaks down via rare regions with nearly uniform $J_P$[37]. Such regions are unlikely in intermediate-scale systems and may also be eliminated in programmable devices, potentially yielding $t_{th} \to \infty$. Below we focus on times below $t_{th}$.

For $U_F = \exp(-iH_1)\exp(-iH_0)$ arising upon perturbing $U_{FO}$ [Eq. (2)], the robustness of TO MBL does not directly follow especially since $H_1$, via $\frac{\pi}{2}\sum_{j\in\gamma_x} X_j$, has couplings comparable to $\delta J$. However, akin to TCs[9,10,56], by $\overline{O}^2 = \mathbb{1}$ we have $U_F^2 \approx \exp[-i(\sum_P 2J_P S_P + \delta H)]$ with $\delta H$ as above, with terms set by those in $H_{0,1}$. Based on this, we expect TO MBL to be robust also for $U_F$. In Supplementary Notes 1, 2, 3, we corroborate this expectation using numerical simulations.

MBL implies that $|\psi_{\mathbf{s},\mathbf{o}}\rangle$ are simultaneous eigenstates of $U_F$ and $T_P = \widetilde{U}S_P\widetilde{U}^\dagger$; these are hence mutually commuting operators. In particular, the $T_P$ are (quasi)local integrals of motion. When $S_P$ corresponds to TO, these $T_P$ are the tLIOMs[35]. When $|\mathbf{s}, o\rangle$ are TO states, then since this cannot change under a local unitary[17,50], so are $|\psi_{\mathbf{s},o}\rangle$; equivalently, tLIOMs imply TO in all eigenstates, which are all still labeled by anyon configurations $\mathbf{s}$[35]. (Conversely, LIOMs[43–47], e.g., via $S_j = Z_j$ or $S_j = Z_jZ_{j+1}$, give topologically trivial states.)

A key extra feature in $U_{FO}$ of Eq. (2) is that $\overline{O}$ specifies the unperturbed eigenbasis in logical space. Hence, not only is $U_{FO}$ TO MBL by the disorder in $J_P$, but the eigenbasis for $U_F$ arising from $U_{FO}$ with local perturbations is formed by eigenstates $|\psi_{\mathbf{s},o}\rangle$ of the $\mathbf{s}$-independent $\widetilde{\mathcal{O}} = \widetilde{U}\overline{O}\widetilde{U}^\dagger$. Since the $T_P$ and $\widetilde{\mathcal{O}}$ form a complete set of integrals of motion, $U_F$ can be expressed using these. The following result implies that this $U_F(\{T_P\}, \widetilde{\mathcal{O}})$ is a TTC, i.e., that $U_{FO}$ of Eq. (2) (and its deformations by local perturbations) satisfies (3) in Definition 1:

**Proposition 1.** If a TO MBL Floquet unitary factorizes as $U_F = \widetilde{\mathcal{O}}e^{-if(\{T_P\})}$ with a (smeared) logical operator $\widetilde{\mathcal{O}}$ and an exponentially local function $f$ of tLIOMs $T_P$, then such a factorization is robust and the system is a TTC with $\widetilde{\mathcal{W}}_\gamma$ in Definition 1 conjugate to $\widetilde{\mathcal{O}}$. Here, exponentially local $f$ means that in

$$f(\{T_P\}) = c_0 + \sum_P c_P T_P + \sum_{P,Q} c_{PQ} T_P T_Q + \cdots, \quad (4)$$

the $c_{PQR...}$ decay exponentially with the largest distance between the centers of the supports of $T_Q$, $T_P$, $T_R$, ....

The proof is given under Methods. Akin to TCs, the structure $U_F = \widetilde{\mathcal{O}}e^{-if(\{T_P\})}$ implies that the Floquet spectrum is organized into eigenstate multiplets with rigid phase patterns[5,7,8]. Specifically, by $o \in \{-1, 1\}$, the spectrum $e^{-i\varepsilon_{\mathbf{s},o}} = oe^{-if(\mathbf{s})}$ displays robust $\pi$ splitting for each $\mathbf{s}$, i.e., within each doublet.

Proposition 1 takes infinite system size $L_\perp$ transversally to the path $\gamma$; otherwise $f$ includes terms proportional to $\widetilde{\mathcal{O}}$ with coefficients decaying exponentially with $L_\perp/\xi$. It is, however, agnostic to the length $L_\parallel$ of the shortest $\gamma$. The TTC, and its $\pi$-paired spectrum, thus emerges exactly even for moderate $L_\parallel$ as long as $L_\perp/\xi \to \infty$. (See also Supplementary Note 1.) At the core of this robustness is $U_F(\{T_P\}, \widetilde{\mathcal{O}})$ not featuring $\widetilde{\mathcal{W}}_\gamma$. This is unlike static TO, with $U_{FO} = e^{-H_0}$ whose logical eigenbasis is arbitrary; here MBL implies only $U_F(\{T_P\}, \widetilde{\mathcal{O}}, \widetilde{\mathcal{W}}_\gamma)$, where $\widetilde{\mathcal{O}}$ and $\widetilde{\mathcal{W}}_\gamma$ enter with coefficients decaying exponentially with $L_\perp$ and $L_\parallel$, respectively, and thus split spectral degeneracies. TTCs are hence, in a sense, even more robust than static TO: they have their own absolute stability.

## Some generalizations

Our TTC construction has a number of immediate generalizations. Firstly, $U_{FO}$ of Eq. (2) can also be defined for systems with $k > 1$ logical qubits (e.g., planar surface codes with multiple holes and/or boundaries alternating several segments with $X$- and $Z$-type boundary checks). We can then choose $\overline{O} = \overline{O}_1\overline{O}_2 \ldots \overline{O}_p$, with logical operators $\overline{O}_j$, for any subset of $p \leqslant k$ logical qubits. In this case, we find TTC signature in each logical operator conjugate to $\overline{O}$ and that each $2^k$-fold multiplet of $H_0$ is $\pi$-split into groups of $2^{k-1}$ levels. Under MBL, the $\pi$ splitting is again absolutely stable, while the $2^{k-1}$-fold degeneracies are as robust as in static TO MBL.

By suitably generalizing the $S_P$ and $\overline{O}$, the TTCs also generalize to any nonchiral Abelian TO[14–16]; these precisely exhaust the TOs compatible with MBL[32–35]. In this way, we can get, e.g., TTC counterparts of ordinary TCs based on any finite Abelian group $G$[6,8].

## Holographic TTC−to−TC correspondence

TTCs also illustrate a dynamical generalization of TO with gapped boundaries: TO with MBL boundaries. Gapped boundaries of (clean) 2D TO have been shown to holographically describe (clean) gapped 1D

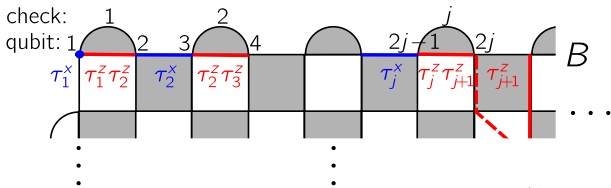

**Fig. 2 | TC ingredients on the TTC boundary $B$.** The boundary checks $S_j = Z_{2j-1}Z_{2j}$ (shown also as red segments) and perturbations $X_1$ (blue dot) and $X_{2j}X_{2j+1}$ (blue segments) become $\tau_j^z\tau_{j+1}^z$ and $\tau_l^x$. The symmetry $P_\tau = \prod_j \tau_j^x$ is a logical $\overline{X}$ of the TO. Also shown is the assignment $\tau_{j+1}^z$ to $\mathcal{Z}_{\gamma_z}$ with $\gamma_z$ ending on qubit $2j$ (dashed) or $2j+1$ (solid vertical line); the two choices differ by a deformation (cf. Fig. 1).

systems, including symmetry broken and symmetry-protected topological phases[57–62]. While the adjective gapped refers to low energies, the correspondence is between operators. It should, therefore, also capture MBL and TCs.

To illustrate this, consider the surface code boundary $B$ in Fig. 2. We start with Eq. (2), with $\overline{O} = \overline{X}$, and add perturbations, first requiring that they commute with $\overline{X}$ and with all $S_P$ except for the $Z$-type checks $S_j = Z_{2j-1}Z_{2j}$ along $B$. Hence, for each bulk **s**, we can focus on the physics at $B$. The allowed perturbations are $\tau_1^x = X_1$ and $\tau_{j+1}^x = X_{2j}X_{2j+1}$ along $B$, and their products with each other and with the $S_P$. To keep perturbations local, we focus on the $\tau_j^x$ and include them in $H_1$. It is suggestive to denote $S_j$, already in $H_0$, as $\tau_j^z\tau_{j+1}^z$ because the $\tau_j^z\tau_{j+1}^z$ and the $\tau_l^x$ satisfy the same relations as if $\tau_j^O$ were Pauli-$O$ operators at site $j$[58]. The $\tau_j^z\tau_{j+1}^z$ and $\tau_l^x$ are precisely the operators for constructing the $\mathbb{Z}_2$ TC $U_F$ via Pauli operators $\tau_j^O$. With the $\mathbb{Z}_2$ symmetry $P_\tau = \prod_j \tau_j^x = \prod_{j \in B} X_j$ being a valid $\overline{X}$ for the TO, and mapping order parameters by identifying $\tau_j^z$ with $\mathcal{Z}_{\gamma_z}$ ending with $Z_{2j-1}$ at $B$, we find that the TTC $U_F$, at boundary $B$, realizes the 1D $\mathbb{Z}_2$ TC. In particular, $T_j = \tilde{U}\tau_j^z\tau_{j+1}^z\tilde{U}^\dagger$ are the LIOMs of the boundary TC. The construction applies also with bulk perturbations, just now with the bulk $S_P$, $\overline{O}$, and $\mathcal{Z}_{\gamma_z}$ replaced by $T_P$, $\tilde{\mathcal{O}}$ and $\tilde{\mathcal{Z}}_{\gamma_z}$. This holographic correspondence is another unifying perspective on TTCs and TCs.

Generalizing this construction, one can also design TO MBL drives that holographically realize other 1D MBL Floquet phases. With suitable nonchiral Abelian TOs, we expect that all 1D symmetry-broken and symmetry-protected topological phases[6,25–27] can arise this way. It will be interesting to see what insights this perspective offers about these 1D Floquet systems.

## Comparisons to some other topological TCs

As mentioned in the Introduction, by their intrinsic 2D TO, which requires no symmetry and supports anyons, our TTCs are distinct from symmetry-protected topological TCs[25–28]; their signatures, via $\widetilde{\mathcal{W}}_\gamma$ (and $\mathcal{W}_\gamma$, see below), are robust to arbitrary (not only symmetry-preserving) perturbations, and originate directly from anyonic statistics. TTCs are also distinct from 1D, including Majorana-based, topological TCs[25,63,64] as these exist a dimension lower and if they have anyon-like particles, these are bound to defects or boundaries.

The closest relation is to intrinsic 2D TO drives that permute anyon species[65,66]. As anyon species are distinguished by the type of $S_P$ (e.g., $X$- and $Z$-type $S_P$ in the surface code), the unperturbed form of such drives replaces $\overline{O}$ in Eq. (2) by a local unitary $\mathcal{A}$ that permutes nearby $S_P$[65,66]. Hence, $\mathcal{A}$ does not commute with the $S_P$, thus the Floquet eigenstates are not $|\mathbf{s}, \mathbf{o}\rangle$ but their superpositions[65,66]. Therefore, while such phases may be robust to local perturbations[65,66], if they are MBL they do not have the tLIOMs $T_P$ we discussed.

## Signatures of TTCs

While $U_F = \tilde{\mathcal{O}}e^{-if(\langle T_P\rangle)}$ implies a TTC, the observable $\widetilde{\mathcal{W}}_\gamma = \tilde{U}\mathcal{W}_\gamma\tilde{U}^\dagger$ in Definition 1 depends on $\tilde{U}$, hence is difficult to access experimentally. It

is easier to access its bare counterpart $\mathcal{W}_\gamma$. We next discuss two signatures in terms of this $\mathcal{W}_\gamma$:

**Proposition 2.** If $U_F$ is an MBL TTC, then (i) in any of its eigenstates $|\psi\rangle$, and for $d(\gamma, \gamma') \gg \xi$ separation, $\langle\psi|\mathcal{W}_\gamma(mT)\mathcal{W}_{\gamma'}(0)|\psi\rangle$ has finite period greater than one as the function of the integer $m$ and decays exponentially with the sum $|\gamma| + |\gamma'|$ of the lengths of $\gamma$ and $\gamma'$, but it does not decay with $d(\gamma, \gamma')$. Furthermore, (ii) this TTC signal emerges also for $\gamma' = \gamma$ with $m \gg 1$ and upon averaging over eigenstates and/or disorder.

We support this claim in Methods using analytic arguments and further substantiate it numerically in Supplementary Notes 2 and 3. This result is a dynamical, time-crystalline, form of the perimeter law, a fundamental feature of TO MBL[32,33], originating in lattice gauge theories[40,41]. In TO MBL, Wilson loops $\mathcal{W}_{\partial A}$ (e.g., a Pauli-$Z$ string equal to the product of all $Z$-type $S_P$ in an area $A$, thus running along the perimeter $\partial A$) have eigenstate expectation $\langle\psi|\mathcal{W}_{\partial A}|\psi\rangle \propto \exp(-\lambda|\partial A|)$ with $\lambda \propto \xi$ (with all lengths in units of lattice spacing). By contrast, in a topologically trivial phase the expectation decays much faster, exponentially with the area $|A|$. In Proposition 2, $A$ is in spacetime, with $\gamma$ (at time $mT$) and $\gamma'$ (at time zero) being its boundaries. The perimeter law has the same status for TO as long-range correlations do for spontaneous symmetry breaking. Hence the TTC perimeter law in Proposition 2 is the TO counterpart of the signatures of TCs' spatiotemporal order.

For the $\mathcal{W}_\gamma$ to give TTC signal of appreciable magnitude as $N \to \infty$, one needs $\gamma$ to have $N$-independent minimal length $L_\parallel$. While this can be achieved via quasi-1D $N \to \infty$ limits, a 2D $N \to \infty$ limit is also possible, e.g., in systems topologically equivalent to a cylinder (such as the example in Fig. 1**b**). We emphasize that keeping $L_\parallel$ finite does not spoil TTC features, as follows from the TTCs' absolute stability.

To detect the TTC perimeter law one must also bypass not having experimental access to eigenstates $|\psi\rangle$. For $\gamma' = \gamma$, one may use quantum typicality[67,68]: using experimentally accessible highly-entangled states[56], one may probe, up to errors exponentially small in $N$, $\langle\psi|\mathcal{W}_\gamma(mT)\mathcal{W}_\gamma(0)|\psi\rangle$ averaged over all eigenstates. One may also use that, for strong MBL (i.e., small $\xi$), $|\psi_{\mathbf{s},\mathbf{o}}\rangle$ deviates only slightly from the experimentally accessible $|\mathbf{s}, \mathbf{o}\rangle$, and thus the desired expectation is approximated by $\langle\mathbf{s}, \mathbf{o}|\mathcal{W}_\gamma[(m+n)T]\mathcal{W}_\gamma(nT)|\mathbf{s}, \mathbf{o}\rangle$ for sufficiently large $n$. Supplementary Notes 2 and 3 include further details and numerics on both approaches. Recent experimental results on our proposed TTCs[42] are promising for the feasibility of detecting the TTC perimeter law.

## TTC in Google Sycamore

The surface code ground states, anyons, and logical operators have seen recent Google Sycamore realizations[49,69] and the same platform has been also used for realizing TCs[56]. (See also Ref. [70] for an IBM realization.) We now describe how Sycamore can be used to create and detect a TTC. We divide Sycamore's square grid of qubits into data qubits and measure qubits (Fig. 3**a**)[21,49]. Data qubits are to be evolved under $U_F$; measure qubits facilitate the desired multi-qubit gates. To generate $e^{-if_P S_P}$ with $Z$-type $S_P$, one may use the standard approach (Fig. 3**b**)[52]. For $X$-type $S_P$, one conjugates the above by $\sqrt{Y}$ on data qubits, using $\sqrt{Y}Z\sqrt{Y}^\dagger = X$. For $\exp(-iH_1)$, one applies suitable single-qubit rotations on data qubits; e.g., using $H_1 = \sum_j [\pi/2 + g_j^{(X)}]X_j + g_j^{(Y)}Y_j + g_j^{(Z)}Z_j$, with $g_j^{(\alpha)} \ll \delta J$ one may study the robustness of TTCs and the TTC perimeter law, as illustrated in Supplementary Notes 2 and 3. All the ingredients, namely the single-qubit rotations, $\sqrt{Y}$, and CNOT are available in Sycamore[48,56]. Detecting TTCs can proceed, e.g., via the interferometric protocol demonstrated in Ref. [69]. The decoherence rates are also compatible with TCs[56] and we expect the same for TTCs.

## Discussion

We have defined TTCs and showed that, combined with MBL, they form a (pre-thermal) dynamical phase. Higher-form symmetries and QEC codes offer complementary ways to link ordinary TCs to TTCs,

**a**

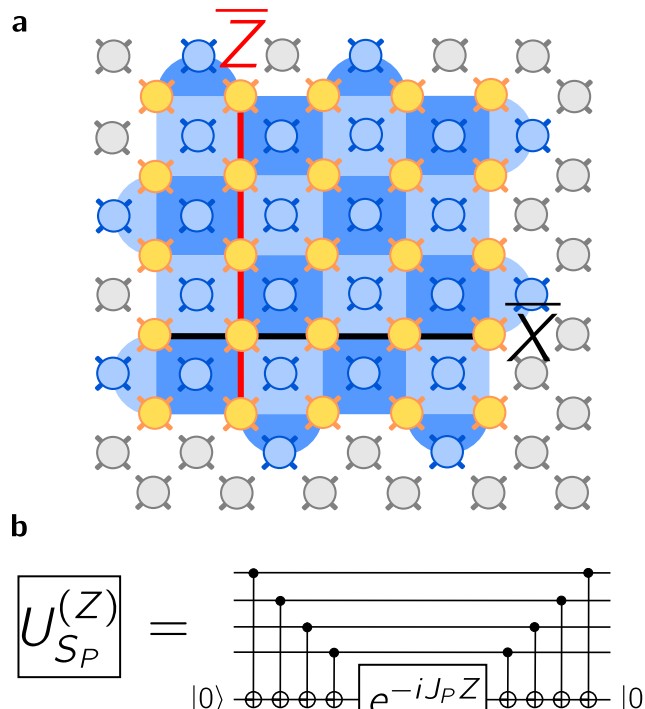

**b**

**Fig. 3 | Key TTC ingredients on the Google Sycamore[49]. a** A Sycamore device, with $N = 25$ data qubits (gold), can realize a system akin to Fig. 1a. (Measure qubits are in blue; the figure is redrawn based on Ref. 49.) Dark (light) blue shaded areas mark $Z$-type ($X$-type) checks $S_P$. **b** The evolution $U_{S_P}^{(Z)} = \exp(-iJ_P \prod_{j \in P} Z_j)$ on four data qubits. The surrounded measure qubit starts in $|0\rangle$; the CNOTs couple to the data qubits. The $U_{S_P}^{(X)}$ evolution arises via conjugating by $\sqrt{Y}$ on the data qubits.

while the holographic correspondence between TTCs and their MBL boundaries offers a reverse link to ordinary TCs. Logical operators serve both as symmetries and as order parameters for TTCs. This leads to interesting interplay, both between spectral pairing patterns and topological degeneracies, and also between MBL and the nonlocality of these operators, the latter manifested in the TTC perimeter law.

The most favorable settings for realizing TTCs feature MBL with short localization lengths (strong MBL). This allows the suppression of finite-size corrections already in moderate-sized systems, and the detection of the TTC perimeter law for an appreciable range of order parameter path lengths. Studying these settings is particularly well suited for programmable quantum processors, such as those featured in the recent, intermediate-scale, demonstration of our proposed TTCs[42], or the Google Sycamore device[48] which also has all the ingredients for creating such a TTC.

## Methods
### Robustness of the TTC drive structure
Here we prove Proposition 1, using considerations analogous to those for establishing TCs' absolute stability[7]. The operator $\widetilde{\mathcal{W}}_\gamma$ below is the one in Definition 1, taken to be conjugate to $\widetilde{\mathcal{O}}$, i.e., $\widetilde{\mathcal{W}}_\gamma \widetilde{\mathcal{O}} = -\widetilde{\mathcal{O}}\widetilde{\mathcal{W}}_\gamma$. The proof hinges on the following: (i) the object $\theta_\gamma = U_F^\dagger \widetilde{\mathcal{W}}_\gamma U_F \widetilde{\mathcal{W}}_\gamma$, illustrated in Fig. 4, is quasilocal transversally to $\gamma$; and (ii) it is independent of the choice of $\gamma$ among the paths deformable into each other.

The transversal quasilocality of $\theta_\gamma$ follows from $U_F$ being a local unitary [it is a finite-time evolution with a local $H(t)$] and from $\widetilde{\mathcal{W}}_\gamma$ being quasilocal transversally to $\gamma$. To show $\theta_\gamma = \theta_{\gamma'}$, we recall that $\mathcal{W}_\gamma$ (along path $\gamma$) can be deformed into $\mathcal{W}_{\gamma'}$ (along path $\gamma'$) by multiplying with a suitable $S_P$ product (cf. Fig. 1). Hence $\widetilde{\mathcal{W}}_\gamma$ can similarly be deformed into $\widetilde{\mathcal{W}}_{\gamma'}$ using a suitable $T_P$ product. Now using the commutation $[T_P, U_F] = [T_P, \mathcal{W}_\gamma] = 0$ and $T_P^2 = 1$, we find $\theta_\gamma = \theta_{\gamma'}$.

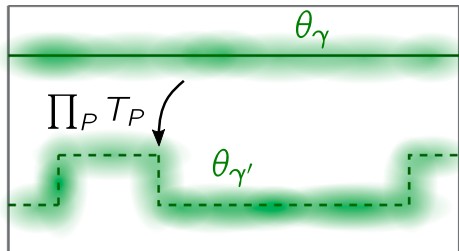

**Fig. 4 | Illustration of the supports of $\theta_\gamma$ and $\theta_{\gamma'}$.** The operators are smeared around the paths $\gamma$ (solid line) and $\gamma'$ (dashed line), respectively. The relation $\theta_\gamma = \theta_{\gamma'}$ that we establish implies $\theta_\gamma \propto \mathbb{1}$, which in turn is shown to imply $\theta_\gamma = \pm \mathbb{1}$.

For separations $d(\gamma, \gamma') \gg \xi$, the path $\gamma$ features in the support of $\theta_{\gamma'}$ only through Pauli strings that contribute to $\theta_{\gamma'}$ with coefficients exponentially small in $d(\gamma, \gamma')/\xi$, and vice versa for $\gamma'$ and $\theta_\gamma$. For this to be consistent with $\theta_\gamma = \theta_{\gamma'}$ we must have $\theta_\gamma = c\mathbb{1}$ (with $c$ a phase, by $\theta_\gamma$ being unitary) up to corrections exponentially small in $L_\perp/\xi$, with $L_\perp$ the system size transversally to $\gamma$. (All our statements below hold up to such corrections.) By $\widetilde{\mathcal{W}}_\gamma^2 = \mathbb{1}$, we have $\theta_\gamma \widetilde{\mathcal{W}}_\gamma = U_F^\dagger \widetilde{\mathcal{W}}_\gamma U_F$, and thus $\theta_\gamma^2 = (U_F^\dagger \widetilde{\mathcal{W}}_\gamma U_F)^2 = \mathbb{1}$. Hence $\theta_\gamma = \pm \mathbb{1}$. This holds for any $U_F(\{T_P\}, \widetilde{\mathcal{O}})$. As perturbations change $\theta_\gamma$ continuously, if $\theta_\gamma = -\mathbb{1}$ then this persists throughout the MBL phase.

The result $\theta_\gamma = \pm \mathbb{1}$ implies that $U_F(\{T_P\}, \widetilde{\mathcal{O}})$ either commutes with $\widetilde{\mathcal{W}}_\gamma$ or anticommutes with it. By $\widetilde{\mathcal{W}}_\gamma \widetilde{\mathcal{O}} = -\widetilde{\mathcal{O}}\widetilde{\mathcal{W}}_\gamma$, $[\widetilde{\mathcal{W}}_\gamma, T_P] = 0$, and $\widetilde{\mathcal{O}}^2 = \mathbb{1}$, we thus find that $U_F$ is either proportional to $\widetilde{\mathcal{O}}$ or independent of $\widetilde{\mathcal{O}}$. The form $U_F = \widetilde{\mathcal{O}}e^{-if(\{T_P\})}$ is the generic structure for $U_F$ proportional to $\widetilde{\mathcal{O}}$, where the exponentially local nature of $f$ follows from the local unitarity of $U_F$. This factorization is thus robust throughout the MBL phase.

For $\theta_\gamma = -\mathbb{1}$, we have $U_F^\dagger \widetilde{\mathcal{W}}_\gamma U_F = -\widetilde{\mathcal{W}}_\gamma$ and hence $\widetilde{\mathcal{W}}_\gamma(mT) = (-1)^m \widetilde{\mathcal{W}}_\gamma$, and thus $\langle\psi|\widetilde{\mathcal{W}}_\gamma(mT)\widetilde{\mathcal{W}}_\gamma(0)|\psi\rangle = (-1)^m$. Considering also the TO the tLIOMs $T_P$ imply in all eigenstates, the system is a TTC.

### TTC perimeter law
We consider, in eigenstate $|\alpha\rangle$, the correlator

$$C_\alpha(m; \gamma, \gamma') = \langle\alpha|\mathcal{W}_\gamma(mT)\mathcal{W}_{\gamma'}|\alpha\rangle$$
$$= \sum_\beta e^{i(\varepsilon_\alpha - \varepsilon_\beta)m}\langle\alpha|\mathcal{W}_\gamma|\beta\rangle\langle\beta|\mathcal{W}_{\gamma'}|\alpha\rangle, \quad (5)$$

where we inserted a Floquet eigenbasis $|\beta\rangle$ with eigenvalues $e^{-i\varepsilon_\beta}$. We first focus on $d(\gamma, \gamma') \gg \xi$ and show that, analogously to TCs[7], in this limit only $|\beta\rangle = |\alpha\rangle$ and $|\beta\rangle = \widetilde{\mathcal{W}}_\gamma|\alpha\rangle \sim \widetilde{\mathcal{W}}_\gamma|\alpha\rangle$ can contribute. To this end, we expand $\mathcal{W}_{\gamma'}$ in terms of operators $T_P, T_P^x, \widetilde{\mathcal{W}}_\gamma, \widetilde{\mathcal{O}}$, where $T_P^x$ anticommutes with $T_P$ (and hence flips its eigenvalue in $|\alpha\rangle$). As $\mathcal{W}_{\gamma'}$ is local transversally to $\gamma^{(\prime)}$, the operator $\widetilde{\mathcal{O}}$ cannot contribute for $L_\perp \gg \xi$ (the limit we assume henceforth). Similarly, $T_P^{(x)}$ can contribute appreciably only if their bare counterparts $S_P^{(x)}$ have their entire support within $\sim \xi$ from $\gamma^{(\prime)}$.

We write $\mathcal{W}_{\gamma^{(\prime)}} = \mathcal{S}_{0\gamma^{(\prime)}} + \mathcal{S}_{1\gamma^{(\prime)}} + \mathcal{S}_{x\gamma^{(\prime)}}$, where $\mathcal{S}_{0\gamma^{(\prime)}}$ is a linear combination of various $\prod_P T_P$, $\mathcal{S}_{1\gamma^{(\prime)}}$ is that of various $\widetilde{\mathcal{W}}_{\gamma^{(\prime)}} \prod_P T_P$, and $\mathcal{S}_{x\gamma^{(\prime)}}$ is similar but with the product featuring at least one $T_P^x$. By the above locality considerations, $\mathcal{W}_{\gamma^{(\prime)}}|\alpha\rangle$ and $|\alpha\rangle$ have the same $T_P$ eigenvalues for $d(P, \gamma^{(\prime)}) \gg \xi$, up to corrections exponentially small in $d(P, \gamma^{(\prime)})/\xi$ (which we neglect henceforth). Hence, $\mathcal{S}_{x\gamma^{(\prime)}}|\alpha\rangle$ is orthogonal to $\mathcal{W}_{\gamma'}|\alpha\rangle$ and vice versa, thus $\mathcal{S}_{x\gamma^{(\prime)}}$ can be neglected in Eq. (5).

The only appreciable contributions are via $\mathcal{S}_{0\gamma^{(\prime)}}$ and $\mathcal{S}_{1\gamma^{(\prime)}}$. Since $\widetilde{\mathcal{W}}_{\gamma^{(\prime)}}$ flips the $\widetilde{\mathcal{O}}$ eigenvalue in $|\alpha\rangle$, there are no cross-terms between $\mathcal{S}_{0\gamma^{(\prime)}}$ and $\mathcal{S}_{1\gamma^{(\prime)}}$. By $U_F^\dagger \mathcal{S}_{0\gamma^{(\prime)}} U_F = \mathcal{S}_{0\gamma^{(\prime)}}$ and $U_F^\dagger \mathcal{S}_{1\gamma} U_F = -\mathcal{S}_{1\gamma}$, we thus get

$$C_\alpha(m; \gamma, \gamma') = c_{0\gamma}^{(\alpha)} c_{0\gamma'}^{(\alpha)} + (-1)^m c_{1\gamma}^{(\alpha)} c_{1\gamma'}^{(\alpha)}, \quad (6)$$

with $c_{j\gamma^{(\prime)}}^{(\alpha)} = \langle \alpha | \widetilde{\mathcal{W}}_{\gamma^{(\prime)}}^{j} \mathcal{S}_{j\gamma^{(\prime)}} | \alpha \rangle$, where we used $\widetilde{\mathcal{W}}_{\gamma^{(\prime)}}^{2} = \mathbb{1}$. The period doubling, in the second term in Eq. (6), is the claimed time-crystalline signature.

As the contributions from $\gamma^{(\prime)}$ factorize in each term, the signal does not decay with $d(\gamma, \gamma')$, as claimed. The perimeter law follows from estimating $|c_{1\gamma^{(\prime)}}^{(\alpha)}|$. To this end, we view the expansion of $\mathcal{W}_{\gamma^{(\prime)}}$ in $T_P$, $T_P^x$, $\widetilde{\mathcal{W}}_\gamma$, $\widetilde{\mathcal{O}}$, as one in an orthogonal basis under the scalar product $\text{Tr}(A^\dagger B)/D_H$ where $D_H$ is the Hilbert space dimension. As MBL provides no structure below the scale $\xi$, all the $\sim 4^{|\gamma^{(\prime)}|/\xi}$ terms (with $|\gamma^{(\prime)}|$, $\xi$ measured in units of lattice spacing) that contribute appreciably to $\mathcal{W}_{\gamma^{(\prime)}}$ have expansion coefficient $a_i$ with roughly the same $|a_i|^2$, with $\sum_i |a_i|^2 = 1$ by $\mathcal{W}_{\gamma^{(\prime)}}^2 = \mathbb{1}$. Hence $|a_i|^2 \sim 4^{-|\gamma^{(\prime)}|/\xi}$. By the Cauchy-Schwarz inequality, $|c_{1\gamma^{(\prime)}}^{(\alpha)}|^2 \leqslant \langle \alpha | \mathcal{S}_{1\gamma^{(\prime)}}^2 | \alpha \rangle$. Due to MBL, no eigenstate $|\alpha\rangle$ is special; hence we can estimate $|c_{1\gamma^{(\prime)}}^{(\alpha)}|^2 \lesssim \text{Tr } \mathcal{S}_{1\gamma^{(\prime)}}^2 / D_H = \sum_i' |a_i|^2$, where $\sum'$ sums over the $\sim 2^{|\gamma^{(\prime)}|/\xi}$ terms that contribute appreciably to $\mathcal{S}_{1\gamma^{(\prime)}}$. Using $|a_i|^2 \sim 4^{-|\gamma^{(\prime)}|/\xi}$, we thus find $|c_{1\gamma^{(\prime)}}^{(\alpha)}| \lesssim 2^{-|\gamma^{(\prime)}|\xi/2}$. The time-crystalline signal thus decays as $\sim 2^{-\xi(|\gamma| + |\gamma'|)/2}$.

We next consider $\gamma = \gamma'$. Now $\mathcal{S}_{x\gamma}$ does not cancel and leads to terms $\sim e^{i(\varepsilon_\alpha - \varepsilon_\beta)m}$, with $|\beta\rangle$ via the $T_P^x$ that can appreciably contribute. For a given $|\alpha\rangle$, these are $D_\gamma \sim 2^{|\gamma|/\xi}$ random phases, each with coefficient $|\langle \alpha | \mathcal{W}_\gamma | \beta \rangle|^2 \sim 1/D_\gamma$, dephasing into a background $\sim 1/\sqrt{D_\gamma}$. For a given $|\alpha\rangle$, this increasingly dominates the $\sim (-1)^m/D_\gamma$ time-crystal signal upon increasing $|\gamma|\xi$.

The eigenstate average

$$C(m; \gamma) = \sum_\alpha C_\alpha(m; \gamma, \gamma)/D_H \tag{7}$$

however, has the time-crystal signal [arising from the $|\beta\rangle$ in Eq. (5) with $e^{i(\varepsilon_\alpha - \varepsilon_\beta)m} = (-1)^m$, present for each $|\alpha\rangle$ by the $\pi$ spectral pairing] preserved at strength $\sim 1/D_\gamma$, but has an increasing number of random phases upon increasing $m$ (i.e., upon resolving splittings from increasingly distant LIOMs' interactions). This leads to a power-law decay[47] of the background towards its long-time magnitude $\sim 1/\sqrt{D_H} = 2^{-N/2}$, well below the $\sim 1/D_\gamma$ time-crystal signal strength. This emergence of the time-crystal signal for $m \gg 1$ is limited by the timescale for resolving exponentially small (in $L_\perp/\xi$) corrections to the $\pi$ spectral pairing and by the thermalization time. The number of random phases can be further increased, and hence the background suppression can be further enhanced, by averaging over disorder.

## Data availability
The datasets for the plots in the Supplementary Information are available upon request.

## Code availability
The codes for our simulations in the Supplementary Information are available upon request.

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

## Acknowledgements

This project was supported by the ERC Starting Grant No. 678795 TopInSy. We also acknowledge support through the Royal Society Research Fellows Enhanced Research Expenses 2021 RF\ERE\210299 and EPSRC ERC underwrite grant EP/X025829/1 (TBW), the Koshland Fellowship at the Weizmann Institute of Science (BH), and the EPSRC grant EP/V062654/1 (BB).

## Author contributions

BB originated the project and conceptualized the QEC, higher-form symmetry, and holographic perspectives. BH and BB developed the unperturbed examples; TBW and BB developed the definitions, propositions, and proofs. TBW performed all numerical work. BB wrote the manuscript with input from all co-authors.

## Competing interests

The authors declare no competing interests.
