## [Transparent Peer Review file · Nature Communications]

Topologically ordered time crystals

Corresponding Author: Professor Benjamin Béri

Version 0:

Reviewer comments:

Reviewer #1

(Remarks to the Author)

In this manuscript, the authors propose topologically ordered time crystals - a prethermal phase of periodically driven systems stabilized via many-body localization. This new dynamical phase features both intrinsic topological order and the spontaneous breaking of discrete time-translation symmetry. The authors describe properties of these topological time crystals such as the subharmonic oscillations of nonlocal order parameters, the robustness against perturbation and the "perimeter law" for the nonlocal order parameters inherited from equilibrium topological order. They put forward examples of topological time crystals based on the surface code Hamiltonian with two different lattice geometries and they briefly discuss the implementation of these models on the Google Sycamore chip.

The paper is written in a very technical language assuming that the reader has a lot of knowledge about many-body localization, discrete time crystals and topological order. As a result, the scope of this paper is limited to experts in these fields.

While the idea of combining topological order and time crystals is new, its relation to other topologically non-trivial time crystals discussed theoretically in Ref. [14] and implemented on a quantum computer in [X. Zhang et al., Nature 607, 468 (2022)] is not sufficiently explained. The level of innovation compared to these existing works thus remains unclear. The clarification of the differences from previously studied SPT order in time crystals is necessary to fully assess this manuscript.

There is a crucial deficiency precluding the publication of this manuscript as a standalone paper. The proposal of topological time crystals is not supported by sufficient evidence that such a new dynamical phase of matter is stable and observable. There is a tension between many-body localization, which typically leads to local conserved quantities, and topological order, which is characterized by nonlocal order parameters. To stabilize time crystals for long times one needs to consider large system sizes. On the other hand, nonlocal order parameters characterizing topological order follow the perimeter law (they decay with increasing length) in the presence of perturbations. The authors propose to resolve this tension by taking nonstandard thermodynamic limits where the length of non-local order parameters is kept finite. The stability of subharmonic response is assumed by the authors. However, it remains unclear whether the time crystals can be stabilized in this nonstandard thermodynamic limit. Moreover, the measurability of subharmonic response in nonlocal order parameters is an unresolved issue. It is thus necessary to further analyze the stability, robustness and measurability of topological time crystals, which could be addressed, for example, by numerical simulations.

Moreover, the robustness to perturbations is not sufficiently studied in this work. The authors present two examples of clean (exactly solvable) models that exhibit subharmonic response. The analysis of robustness against perturbations, which is the key feature of time crystals, is crucially missing for these examples. This could be also addressed by numerical simulations.

Furthermore, we have several scientific and technical questions that should be addressed before the publication of this manuscript in any journal:

- 1) The paragraph starting on line 142 is very hard to follow.
- 2) In Proposition 2, the statement "...the thermodynamic limit keeps the length of γ finite." is unclear.
- 3) In section "Holographic TTC-to-TC correspondence", the authors contrast the topological time crystal with a system where topological local integrals of motion anti-commute with the nonlocal order parameters. However, it is unclear how such a system comes about. A concrete example of such a system would help to understand it.
- 4) Line 340: "For exp $(-i H_0)$, one applies phase gates on data qubits." Since H_0 is the sum of Pauli X operators with the prefactor $\pi/2$, we do not see how this is implemented via phase gates. Instead, this unitary can be implemented as X gates

on data qubits.

5) The formulations of Proposition I in the main text and in the supplementary material are inconsistent.

In summary, this manuscript presents new ideas arising from the combination of time crystals and topological order. However, the proposal is not supported by a sufficient analysis and key questions about the stability, robustness and measurability of proposed topological time crystals remain unclear. For these reasons, the manuscript does, in our opinion, not reach the high standards required for publication in Nature Communications.

Reviewer #2

(Remarks to the Author)

Reviewer #3

(Remarks to the Author)

Wahl et al. theoretically define a notion of time crystalline topological order and explore the basic properties of such nonequilibrium phases from several perspectives. The primary observable signature of these topological time crystals (TTC) is a subharmonic response of the expectation value of a Wilson loop (logical operator) expectation value.

The TTC phases proposed by the authors is theoretically well defined and their analysis of the TTC properties is varied and, to my knowledge, correct. That these phases can be realized in near-term experiments makes this article of broad interest to researchers involved with quantum dynamics or topological phases. The following comments and questions should be addressed before publication.

1. My main criticism is that several sections of the paper are quite terse, and will only be of use to experts to whom the point of the section is obvious. For instance, the "Holographic TTC-to-TC correspondence" section is quite schematic, and does not provide enough detail for a non-expert to appreciate the point being made. This section should be expanded, and perhaps illustrated with an explanatory figure. At least, some more detailed discussion should be included in the supplemental material.

2. The authors should explain the position of the TTC phase within the broader theory landscape in more detail. Specifically:
2A. At the end of the "Comparison with other TCs" section, the authors assert that TTCs, as defined in this article, are distinct from several other topological periodically driven systems. It would be helpful for there to be more discussion on exactly what these differences are, and whether there are less direct relationships between the various phases mentioned.
2B. The article is primarily concerned with Z2 topological order, with a brief comment being made about Zn. Is this the complete class of topological orders for which a notion of TTC can be defined? It seems like any topological order compatible with localization should be admissible.

3. This article spent a long time on the arXiv. A few weeks after it was first posted in May 2021, [Bomantara, Phys. Rev. B 104, 064302 (2021), arXiv:2106.01748] appeared, and was published in PRB later the same year. This paper studies the same phase which is identified by Wahl et al., with an emphasis on different features. While these works were clearly simultaneous and independent, the existence of this related work should be acknowledged in the present article.

4. The authors make the parenthetical comment "We can also get [an infinite thermalization time] by eliminating rare weak-disorder regions [34], e.g., by using quasi-periodic potentials". This statement is speculative, and not a conclusion of Ref. [34]. While MBL with random disorder is believed to be ultimately unstable in dimensions higher than 1 (due to rare regions), the converse statement that systems with non-random disorder---and no rare regions---exhibit a stable MBL phase in higher dimensions is not known to be true.

5. The requirement at the top of page 4 that the logical Z operator has odd length seems to be a consequence of the choice of model, and not a fundamental requirement of the phase, which should be clarified. Taking the sum in H0 to be restricted to a line transversal to ZL will make the Floquet unitary close to a nontrivial logical operator for any geometry.

Version 1:

Reviewer comments:

Reviewer #1

(Remarks to the Author)

Reviewer #2

(Remarks to the Author)

I co-reviewed this manuscript with one of the reviewers who provided the listed reports. This is part of the Nature

Communications initiative to facilitate training in peer review and to provide appropriate recognition for Early Career Researchers who co-review manuscripts.

Reviewer #3

(Remarks to the Author)

The substantial changes made to the manuscript have addressed my concerns from the previous report to my satisfaction. Overall, I find the presentation to be significantly clearer than the previous draft.

RESPONSE TO REVIEWERS #1 AND #2

We thank the reviewers for their careful reading of our manuscript and for their report. The reviewers' main concerns are (i) the too technical presentation of our work, preventing it from being accessible to nonexperts, and (ii) the incompleteness of our analysis of the robustness and observability of TTCs, which they suggest to remedy via numerical simulations.

We appreciate the constructive suggestions and comments. We have rewritten the paper so that our research questions, results, and their relation to previous work are better conveyed already in the introduction, and we made sure that the technical concepts and ingredients are introduced gradually. We trust that this substantially revised new version is accessible to a broad audience. We have also reinforced our analysis of the robustness and observability of TTCs, with the new additions including numerical simulations.

In the following, we address the reviewers' comments and questions (which we number for clarity), and indicate the corresponding changes.

1. "The paper is written in a very technical language assuming that the reader has a lot of knowledge about many-body localization, discrete time crystals and topological order. As a result, the scope of this paper is limited to experts in these fields."

Response 1. We thank the reviewers for highlighting the technical nature of the presentation. On reflection we agree with this assessment of the initially submitted version of the paper.

Revision 1. We have implemented an essentially complete rewrite of the manuscript. The new version includes a brief introduction to time crystals and topological order (p2) as well as to many-body localization and its interplay with topological order (p3). These parts review the essential ingredients, in a manner accessible to nonexperts, before the paper would discuss the new ideas and results.

2. "While the idea of combining topological order and time crystals is new, its relation to other topologically non-trivial time crystals discussed theoretically in Ref. [14] and implemented on a quantum computer in [X. Zhang et al., Nature 607, 468 (2022)] is not sufficiently explained. The level of innovation compared to these existing works thus remains unclear. The clarification of the differences from previously studied SPT order in time crystals is necessary to fully assess this manuscript."

Response 2. We thank the reviewers for prompting us to better contrast our work with these seminal papers. As the reviewers noted, the key difference is that these works feature symmetry-protected topological (SPT) order, which requires symmetries (beyond discrete time-translation) for its existence. The topologically ordered time crystals we define and study require no symmetry. They can, however, be viewed on the same footing with conventional time crystals, upon invoking modern notions of symmetry (in particular 1-form global symmetries). Furthermore, unlike Floquet SPT phases, the TTCs have anyonic excitations and spectral multiplet structure that depends on the topology of the configuration space.

Revision 2. We now organize our introduction (and our abstract) around our discovery of time crystals arising from a "quantum fabric" that is not reliant on symmetries, but is based on a distinct type of quantum order, namely topological order (of the intrinsic kind, with anyonic quasiparticles and topology-dependent spectral multiplets). In addition to [14] (now [25]), we now also cite X. Zhang et al., Nature 607, 468 (2022) (Ref. 28).

3. "The proposal of topological time crystals is not supported by sufficient evidence that such a new dynamical phase of matter is stable and observable. There is a tension between many-body localization, which typically leads to local conserved quantities, and topological order, which is characterized by nonlocal order parameters. To stabilize time

48 crystals for long times one needs to consider large system sizes. On the other hand,
 49 nonlocal order parameters characterizing topological order follow the perimeter law (they
 50 decay with increasing length) in the presence of perturbations. The authors propose to
 51 resolve this tension by taking nonstandard thermodynamic limits where the length of
 52 non-local order parameters is kept finite.”

53 **3a.** “The stability of subharmonic response is assumed by the authors. However, it
 54 remains unclear whether the time crystals can be stabilized in this nonstandard thermo-
 55 dynamic limit. (...) Moreover, the robustness to perturbations is not sufficiently studied
 56 in this work. The authors present two examples of clean (exactly solvable) models that
 57 exhibit subharmonic response. The analysis of robustness against perturbations, which is
 58 the key feature of time crystals, is crucially missing for these examples.”

59 **3b.** “Moreover, the measurability of subharmonic response in nonlocal order parameters
 60 is an unresolved issue.”

61 “It is thus necessary to further analyze the stability, robustness and measurability of topo-
 62 logical time crystals, which could be addressed, for example, by numerical simulations.”

63 **Response 3a.** We agree with the reviewers that the assumptions under which our analytical
 64 arguments guarantee the robustness of TTCs needed to be better elucidated, and the validity of
 65 these assumptions needed further substantiation.

66 In short, the robustness of TTCs, *provided the robustness of the underlying TO MBL* (in the
 67 pre-thermal sense appropriate for 2D MBL systems), follows from the robustness of the TTC drive
 68 structure (Proposition 1.). The core of establishing this is establishing that $\theta_\gamma = U_F^\dagger \widetilde{\mathcal{W}}_\gamma U_F \widetilde{\mathcal{W}}_\gamma = \pm \mathbb{1}$
 69 (with $\widetilde{\mathcal{W}}_\gamma$ conjugate to $\widetilde{\mathcal{O}}$ in Proposition 1.) up to corrections exponentially small in the system size
 70 L_\perp transversally to the path γ . This implies the robustness of TTCs under generic local perturbations.
 71 As we now emphasize in our discussion of Proposition 1., the requirement is only that L_\perp be large,
 72 however Proposition 1. is agnostic to the minimum length L_\parallel the path γ can have. Hence it holds
 73 equally well for a thermodynamic limit where L_\parallel is kept finite. As we now better emphasize, this
 74 corresponds to a TTC incarnation of the absolute stability in TCs [1].

75 We however agree that due to the proviso, namely the robustness of TO MBL, the robustness of
 76 TTCs needed substantiation. We now do this using two considerations:

77 The first relates the robustness of TTCs to that of static TO. This invokes an observation analogous
 78 to that used to argue for the robustness of TCs [2, 3], namely that for a perturbed TTC drive U_F we
 79 have $U_F^2 \approx \exp[-i(2H_0 + \delta H)]$ with $H_0 = \sum_P J_P S_P$ being an unperturbed TO MBL Hamiltonian and
 80 δH featuring local perturbations. The pre-thermal TO MBL of H_0 is expected, and was numerically
 81 observed [4], to be robust against δH for J_P disordered and δH featuring only local couplings with
 82 coefficients of order $g \ll \delta J$, where δJ is the width of the J_P distribution [5–8]. Given that U_F^2 and
 83 U_F have the same eigenbasis, we expect TO MBL to be robust also for U_F . We certainly expect this
 84 for systems with standard thermodynamic limit (e.g., square geometry) but also if we, e.g., remove
 85 some plaquettes from the bulk of the system, thus introducing a hole such that the system can have
 86 finite L_\parallel in the thermodynamic limit. We also expect the same to hold in quasi-1D systems, but we
 87 are not aware of numerical support for this in the literature.

88 Our second consideration therefore supports the robustness of TTCs by direct numerical simulations
 89 of surface code TTCs. We study systems both on the square geometry (up to $5 \times 5 = 25$ qubits) and
 90 in quasi-1D settings (up to systems with 3×8 qubits) and the results are consistent with the TTC
 91 phase being robust against local perturbations.

92 **Revision 3a.** We have introduced a section titled “Robustness of TTCs via MBL”. This includes
 93 a discussion of the robustness of TO MBL in the first paragraphs leading up to Proposition 1. After
 94 Proposition 1., we also emphasize that large L_\parallel is not a requirement, and highlight that this implies
 95 a TTC form of absolute stability. We now prove Proposition 1. in the Methods section. We present
 96 the surface code TTC simulations in the Supplementary Information.

97 **Response 3b.** We agree that the measurability of the TTC signal via \mathcal{W}_γ deserved further
 98 discussion. Regarding the challenges directly due to the nonlocality of \mathcal{W}_γ , the key ingredient of
 99 the interferometric protocol we refer to in the main text (in section “TTC in Google Sycamore”) is
 100 a controlled $|\gamma\rangle$ -qubit Z gate, where $|\gamma\rangle$ is the length of γ (understood as the number of qubits in
 101 γ). The demonstration of surface-code anyon braiding in the Google Sycamore [9] used controlled

6-qubit Pauli operators. We therefore believe that using \mathcal{W}_γ up to at least $|\gamma| = 6$ should be feasible. The measurability of the TTC signal also requires the signal to remain appreciable with small perturbations to the system. (In the unperturbed system, the signal has unit magnitude.) Our numerical simulations in the Supplementary Information confirm this to be the case: for perturbations of strength $g = 0.2$ (corresponding to $0.2/\pi \approx 0.064$ of the strength of dominant couplings), the TTC signal has magnitude ~ 0.5 for $|\gamma| = 7$. The results from the recent experiment [10] realizing our proposal further support the observability of the TTC signal via \mathcal{W}_γ .

Revision 3b. We have reworded Proposition 2. from an “only if” statement to a more explicit statement of the signatures corresponding to the TTC perimeter law and expanded it to also include autocorrelators. We now provide our arguments supporting Proposition 2. in the Methods section. Under Proposition 2., we have added discussions of experimental protocols to detect TTCs. The Supplementary Information includes further discussions about these protocols (including the feasibility of the controlled $|\gamma|$ -qubit Z) and our numerical simulations therein illustrate the resulting signatures.

“Furthermore, we have several scientific and technical questions that should be addressed before the publication of this manuscript in any journal:”

T1. “The paragraph starting on line 142 is very hard to follow.”

Response and revision T1. We thank the reviewers for highlighting this. In the paragraphs following Proposition 1., we now focus on the key implications of the proposition. We now include the (revised version of the) proof of Proposition 1. in Methods.

T2. “In Proposition 2, the statement ‘...the thermodynamic limit keeps the length of γ finite.’ is unclear.”

Response and revision T2. We agree that the formulation of Proposition 2. was far from ideal. We have now reformulated it to better convey the signatures corresponding to the TTC perimeter law (see also Revision 3b.).

T3. “In section ‘Holographic TTC-to-TC correspondence’, the authors contrast the topological time crystal with a system where topological local integrals of motion anti-commute with the nonlocal order parameters. However, it is unclear how such a system comes about. A concrete example of such a system would help to understand it.”

Response and revision T3. We were encouraged to expand on this section by all reviewers and we agree that this benefits the paper. We now provide a concrete example, illustrated by a new figure, of such TC at a 1D boundary of a TTC.

T4. “Line 340: ‘For $\exp(-iH_0)$, one applies phase gates on data qubits.’ Since H_0 is the sum of Pauli X operators with the prefactor $\pi/2$, we do not see how this is implemented via phase gates. Instead, this unitary can be implemented as X gates on data qubits.”

Response and revision T4. We agree that the term “phase gate” was not right—while we used it to refer to $\exp(-ig_j X_j)$ we realize that it usually refers to rotations about Z , sometimes even to the S gate. We have now reworded the corresponding section, using $H_1 = \sum_j [\pi/2 + g_j^{(X)}] X_j + g_j^{(Y)} Y_j + g_j^{(Z)} Z_j$ and calling the corresponding gates single-qubit gates. Since generic single-qubit X and Z rotations have already been demonstrated in the Sycamore (they were, e.g., essential ingredients in the Sycamore TC experiment [11]), any single-qubit gate is also available, by the Euler angle parameterization of $SU(2)$.

T5. “The formulations of Proposition I in the main text and in the supplementary material are inconsistent.”

Response and revision T5. We thank the reviewers for pointing this out. While the difference was intentional, to account for the seemingly more general situation of finite L_\parallel in the Supplementary Information, we now state Proposition 1. only in the main text and explain large L_\parallel not being a requirement in the paragraphs following Proposition 1.

RESPONSE TO REVIEWER #3

149

150 We thank the reviewer for their careful reading of our manuscript and for their favourable assess-
151 ment that

152 “The TTC phases proposed by the authors is theoretically well defined and their analysis
153 of the TTC properties is varied and, to my knowledge, correct. That these phases can
154 be realized in near-term experiments makes this article of broad interest to researchers
155 involved with quantum dynamics or topological phases.”

156 The reviewer’s main concern is the too terse presentation of some of the ideas, preventing the paper
157 from being accessible to nonexperts. On reflection, we agree with this assessment. To make the paper
158 broadly accessible, we have implemented an entire rewrite of the manuscript. As noted also in our
159 response to Reviewers #1 and #2, our research questions, results, and their relation to previous work
160 are now better conveyed already in the introduction, and we made sure that the technical concepts
161 and ingredients are introduced gradually. The revised version also addresses all of the reviewer’s
162 other comments and questions, as we discuss below.

163 **1.** “My main criticism is that several sections of the paper are quite terse, and will
164 only be of use to experts to whom the point of the section is obvious. For instance, the
165 ‘Holographic TTC-to-TC correspondence’ section is quite schematic, and does not provide
166 enough detail for a non-expert to appreciate the point being made. This section should
167 be expanded, and perhaps illustrated with an explanatory figure. At least, some more
168 detailed discussion should be included in the supplemental material.”

169 **Response and revision 1.** We thank the reviewer for prompting us to make the paper more
170 accessible, and for their constructive suggestions on how this may be achieved. We have implemented
171 an essentially complete rewrite of the paper. As part of this, we have expanded the ‘Holographic
172 TTC-to-TC correspondence’ section. The changes here include making the construction more explicit
173 using the surface code as an example and supporting the discussion with an explanatory figure.

174 **2.** “The authors should explain the position of the TTC phase within the broader theory
175 landscape in more detail. Specifically:”

176 **2A.** “At the end of the ‘Comparison with other TCs’ section, the authors assert that
177 TTCs, as defined in this article, are distinct from several other topological periodically
178 driven systems. It would be helpful for there to be more discussion on exactly what these
179 differences are, and whether there are less direct relationships between the various phases
180 mentioned.”

181 **2B.** “The article is primarily concerned with \mathbb{Z}_2 topological order, with a brief comment
182 being made about \mathbb{Z}_n . Is this the complete class of topological orders for which a notion
183 of TTC can be defined? It seems like any topological order compatible with localization
184 should be admissible.”

185 **Response and revision 2A.** We thank the reviewer for highlighting that this needed improve-
186 ment. The paper now makes several comparisons. Firstly, as part of the introduction, we now make a
187 clear distinction from symmetry-protected topological Floquet systems: while these need symmetries
188 (beyond discrete time translation) for their existence, TTCs build on topological order (TO) and this
189 requires no symmetry.

190 In the section ‘Comparisons to some other topological TCs’, in addition to reminding of this, we
191 further distinguish from 1D topological TCs: in addition to these existing a dimension lower, when
192 these are topological in the sense of supporting anyon-like particles, these anyons are always bound
193 to some defects or boundaries, unlike the 2D TO we study where they exist as deconfined excitations.
194 Finally, in the same section, we now also expand on the comparison to drives U_F that permute anyon
195 species. These systems also build on intrinsic TO. However, as we note, if they can be stabilized by
196 MBL, they cannot have the T_P (local-unitary-dressed stabilizers) as local integrals of motion since
197 the P label anyons, thus the anyon permutation by U_F precludes U_F from commuting with the T_P .

198 In the section ‘Holographic TTC-to-TC correspondence’ we also include a brief comment on a
199 less direct relationship: with suitably chosen nonchiral Abelian TO, namely the quantum double

200 $\mathcal{D}(G)$ (related to G -gauge theory) with finite Abelian G , all 1D Floquet MBL phases, including
 201 all symmetry-protected topological phases with symmetry group G , can be realized on the TTC
 202 boundary. One of us has work in progress confirming this, but in the paper we phrased it only as an
 203 expectation (and omitted the technical details).

204 **Response and revision 2B.** The reviewer is right that \mathbb{Z}_n was just an example and not the most
 205 general TO compatible with TTCs. Indeed, any nonchiral Abelian TO is compatible with MBL and
 206 thus with TTCs. We now better emphasize this. We now mention such TOs being required for MBL
 207 already in the introduction, and explicitly note that our construction generalizes to all such TOs in
 208 the section ‘Some generalizations’.

209 **3.** “This article spent a long time on the arXiv. A few weeks after it was first posted in
 210 May 2021, [Bomantara, Phys. Rev. B 104, 064302 (2021), arXiv:2106.01748] appeared,
 211 and was published in PRB later the same year. This paper studies the same phase which
 212 is identified by Wahl et al., with an emphasis on different features. While these works
 213 were clearly simultaneous and independent, the existence of this related work should be
 214 acknowledged in the present article.”

215 **Response and revision 3.** The reviewer is completely right that this should be acknowledged.
 216 We now refer to this work (our Ref. 54) when noting, under Eq. 2, the existence of similar surface
 217 code constructions. (The other two references there, Refs. 7 and 8, precede our work and in both in
 218 the previous and the current version we cite them in the introduction.)

219 **4.** “The authors make the parenthetical comment ‘We can also get [an infinite thermal-
 220 ization time] by eliminating rare weak-disorder regions [34], e.g., by using quasi-periodic
 221 potentials’. This statement is speculative, and not a conclusion of Ref. [34]. While MBL
 222 with random disorder is believed to be ultimately unstable in dimensions higher than 1
 223 (due to rare regions), the converse statement that systems with non-random disorder—
 224 and no rare regions—exhibit a stable MBL phase in higher dimensions is not known to
 225 be true.”

226 **Response and revision 4.** We agree that we should have phrased that comment more carefully—
 227 it indeed is a speculative idea and we agree that the converse statement does not follow from known
 228 results. We now word that comment such that its speculative nature should be apparent: we write
 229 “Such [rare] regions are unlikely in intermediate-scale systems and may also be eliminated in pro-
 230 grammable quantum devices, potentially yielding $t_{\text{th}} \rightarrow \infty$.”

231 **5.** “The requirement at the top of page 4 that the logical Z operator has odd length
 232 seems to be a consequence of the choice of model, and not a fundamental requirement
 233 of the phase, which should be clarified. Taking the sum in H_0 to be restricted to a line
 234 transversal to Z_L will make the Floquet unitary close to a nontrivial logical operator for
 235 any geometry.”

236 **Response and revision 5.** The reviewer is right. We now set up the TTC discussion with this
 237 approach (Eq. 2, and the preceding discussion introducing logical operators \bar{O} as line operators) and
 238 comment only later, after Eq. 3, on ways to eliminate the path choice in \bar{O} . The manifestly path-
 239 choice-free approach via a uniform drive Hamiltonian, which is guaranteed to yield a TTC under
 240 conditions that include an odd-length logical Z , is noted as a particular example.

-
- 241 [1] C. W. von Keyserlingk, V. Khemani, and S. L. Sondhi, Absolute stability and spatiotemporal long-range
 242 order in Floquet systems, *Phys. Rev. B* **94**, 085112 (2016).
 243 [2] N. Y. Yao, A. C. Potter, I.-D. Potirniche, and A. Vishwanath, Discrete time crystals: Rigidity, criticality,
 244 and realizations, *Phys. Rev. Lett.* **118**, 030401 (2017).
 245 [3] D. V. Else, B. Bauer, and C. Nayak, Prethermal phases of matter protected by time-translation symmetry,
 246 *Phys. Rev. X* **7**, 011026 (2017).

- 247 [4] F. Venn, T. B. Wahl, and B. Béri, Many-body-localization protection of eigenstate topological order in
248 two dimensions, arXiv:2212.09775.
- 249 [5] D. A. Huse, R. Nandkishore, V. Oganesyan, A. Pal, and S. L. Sondhi, Localization-protected quantum
250 order, *Phys. Rev. B* **88**, 014206 (2013).
- 251 [6] B. Bauer and C. Nayak, Area laws in a many-body localized state and its implications for topological
252 order, *J. Stat. Mech.* **2013**, P09005 (2013).
- 253 [7] A. C. Potter and A. Vishwanath, Protection of topological order by symmetry and many-body localiza-
254 tion, arXiv:1506.00592.
- 255 [8] T. B. Wahl and B. Béri, Local integrals of motion for topologically ordered many-body localized systems,
256 *Phys. Rev. Res.* **2**, 033099 (2020).
- 257 [9] K. J. Satzinger, Y. Liu, A. Smith, C. Knapp, M. Newman, C. Jones, *et al.*, Realizing topologically
258 ordered states on a quantum processor, *Science* **374**, 1237 (2021).
- 259 [10] L. Xiang *et al.*, Long-lived topological time-crystalline order on a quantum processor, arXiv:2401.04333.
- 260 [11] X. Mi *et al.*, Time-crystalline eigenstate order on a quantum processor, *Nature* **601**, 531 (2022).

2 **RESPONSE TO REVIEWERS #1 AND #2**

3 We thank the reviewers for their careful reading of our manuscript and our responses, and their
4 favorable assessment of our revisions, including the generality of our results. We also appreciate the
5 constructive comments which have helped us further improve our manuscript.

6 In the following, we address the comments and indicate the corresponding changes. We start with
7 the main comments (which we label as A and B), and then address the minor points (1, 2, 3).

8 **A.** “The authors clarify in the revised manuscript that topologically ordered time crystals
9 emerge from the topological order of the underlying Hamiltonians and that, in contrast
10 to existing Floquet symmetry-protected topological phases [25-28], do not require global
11 symmetries. However, it remains partially unclear what the new features of this new
12 phase of matter are in contrast to the existing topological time crystals. The current
13 manuscript relies on the reader to deduce the differences from existing phases themselves.
14 It would be helpful to explicitly describe differences in the signatures of the topologically
15 ordered and symmetry-protected time crystals, for example in section “Comparisons to
16 some other topological time crystals”. ”

17 **Response and revision A.** We thank the reviewers for their suggestion. We now also emphasize
18 that the signatures are robust to arbitrary (not only symmetry-preserving) perturbations and that
19 they originate directly from anyonic statistics (see also our response to point B).

20 **B.** “In this context, the authors mention that topologically ordered systems exhibit any-
21 onic excitations. While it is well-known that low energy excitations of topologically or-
22 dered systems feature anyonic statistics, the role of such anyonic excitations is not ex-
23 plained in this manuscript considering driven systems. It would be helpful to explain how
24 such low energy physics emerges in these driven systems.”

25 **Response and revision B.** We thank the reviewers for this comment as well; we agree that
26 addressing this helps better convey the conceptual ideas. We have made a number of changes:

27 Firstly, we now note, above Eq. 2, what the various QEC objects correspond to in terms of anyon
28 theories. Specifically, that “the S_P eigenvalues label anyon configurations, a logical operator \mathcal{W}_γ drags
29 a corresponding anyon along its path γ , and its algebra $\bar{O}^\dagger \mathcal{W}_\gamma \bar{O} = e^{2i\theta_{ow}} \mathcal{W}_\gamma$ with its conjugate \bar{O}
30 encodes mutual statistics through the angle θ_{ow} .” (At this stage, this highlights that each eigenstate,
31 not only low-lying ones, is labeled by anyon configurations in the unperturbed limit.)

32 Secondly, still in the discussion of the unperturbed model, we now highlight under Eq. 3 that the
33 time crystalline signal originates from the mutual semion statistics of surface code anyons. (This
34 feature and its MBL counterpart also relate to our response to point A.)

35 Further, under Definition 1, we now note that the anyon interpretation continues to hold for TO
36 MBL systems for the corresponding smeared operators.

37 Finally, when discussing TO MBL eigenstates (in the second paragraph above Proposition 1), we
38 note that these are still labeled by anyon configurations. (Hence under TO MBL anyons remain
39 meaningful for all eigenstates, similarly to the unperturbed limit.)

40 **1.** “ On line 22 the term “Quantum orders” is colloquial. A more precise term would be
41 appropriate to avoid misunderstanding.”

42 **Response and revision 1.** We thank the reviewers for this note. We now write “Many-body
43 systems can however display orders other than symmetry breaking.”

44 **2.** “It is unclear how the approximate expression for the unitary U_F on line 247 is derived.
 45 It is also unclear if δH in this expression is related to δH in the previous paragraph. The
 46 authors refer the reader to the supplementary information but there is no derivation of
 47 this expression either. It would be helpful to add this derivation.”

48 **Response and revision 2.** We thank the reviewers for raising these points for clarification. For
 49 a static Hamiltonian, we use δH to describe the class of perturbations under which a TO MBL phase
 50 can persist (in the pre-thermal sense). We have now clarified the wording of this paragraph to make
 51 this intended meaning clear by writing “...(pre-thermal) TO MBL is expected for any δH with local
 52 terms with couplings of order $g \ll \delta J$ ”. To clarify how δH in the paragraph with the approximate
 53 expression for U_F^2 relates to this, we now note that the δH there is as above (i.e., has the above
 54 mentioned features required to preserve TO MBL). We have also added a new Supplementary Note
 55 that includes the derivation for $U_F^2 \approx \exp[-i(\sum_P 2J_P S_P + \delta H)]$.

56 **3.** “The discussion of the Floquet spectrum starting on line 283 is very technical and hard
 57 to follow. It would be helpful to expand this discussion, perhaps in the supplementary
 58 information, and plot the Floquet spectrum in a figure.”

59 **Response and revision 3.** We thank the reviewers for pointing this out. We have expanded
 60 this discussion in the new Supplementary Note referred to in point 2. We have also added new
 61 numerical results demonstrating the TTC absolute stability by showing that the π -pairing receives
 62 only exponentially decaying corrections where the exponential decay is with L_\perp even for $L_\perp > L_\parallel$.

63 RESPONSE TO REVIEWER #3

64 We thank the reviewer for their careful reading of our revision and responses and for their favorable
 65 assessment of our revised manuscript. We are very happy to hear that the presentation became
 66 significantly clearer with our revision.